# Investigation of the Effect of C-Terminal Adjacent Phenylalanine Residues on Asparagine Deamidation by Quantum Chemical Calculations

**DOI:** 10.3390/ijms26146819

**Published:** 2025-07-16

**Authors:** Koichi Kato, Haruka Asai, Tomoki Nakayoshi, Ayato Mizuno, Akifumi Oda, Yoshinobu Ishikawa

**Affiliations:** 1Faculty of Pharmaceutical Sciences, Shonan University of Medical Sciences, 16-10 Kamishinano, Totsuka-ku, Yokohama 244-0806, Japan; yoshinobu.ishikawa@sums.ac.jp; 2Faculty of Pharmacy, Meijo University, 150 Yagotoyama, Tempaku-ku, Nagoya 468-8503, Japan; nakayosi@meijo-u.ac.jp (T.N.); 254331505@ccmailg.meijo-u.ac.jp (A.M.); oda@meijo-u.ac.jp (A.O.); 3Department of Pharmacology, School of Medicine, Aichi Medical University, 1-1 Yazakokarimata, Nagakute 480-1195, Japan; asai.haruka.406@mail.aichi-med-u.ac.jp; 4Institute for Advanced Research, Nagoya University, Furo-cho, Chikusa-ku, Nagoya 464-0814, Japan; 5Institute for Protein Research, Osaka University, 3-2 Yamadaoka, Suita 565-0871, Japan

**Keywords:** deamidation, post-translational modification, cataract, crystallin, quantum chemical calculation

## Abstract

The deamidation rate is relatively high for Asn residues with Phe as the C-terminal adjacent residue in γS-crystallin, which is one of the human crystalline lens proteins. However, peptide-based experiments indicated that bulky amino acid residues on the C-terminal side impaired Asn deamination. In this study, we hypothesized that the side chain of Phe affects the Asn deamidation rate and investigated the succinimide formation process using quantum chemical calculations. The B3LYP density functional theory was used to obtain optimized geometries of energy minima and transition states, and MP2 and M06-2X calculations were used to obtain the single-point energy. Activation barriers and rate-determining step changed depending on the orientation of the Phe side chain. In pathways where an interaction occurred between the benzene ring and the amide group of the Asn residue, the activation barrier was lower than in pathways where this interaction did not occur. Since the aromatic ring is oriented toward the Asn side in experimentally determined structures of γS-crystallin, the above interaction is considered to enhance the Asn deamidation.

## 1. Introduction

Crystallins are soluble proteins that are abundant in the mammalian ocular lens and are composed of various subtypes [1,2,3]. The lifespan of human crystallins in lens fiber cells is remarkably long, often exceeding 90 years, and the solubility and stability of the native three-dimensional (3D) structure are necessary to maintain lens transparency and refractive properties [4,5,6]. Various subtypes of crystalline were reported, of which α-crystallins are members of the heat-shock protein (Hsp) family, while β- and γ-crystallins are unrelated to Hsps [1,2,3]. The β- and γ-crystallins are structurally homologous superfamilies, with a monomeric mass of about 20 kDa. The structure of β- and γ-crystallin consists of two domains containing two Greek-key β-sheet motifs, each linked by a short connecting peptide. Crystallin aggregation is the etiology of cataracts, the most common cause of blindness [7]. It has been suggested that non-enzymatic deamidation of Asn residues on crystallins leads to abnormal protein-protein interactions and aggregation, causing cataracts [8,9,10,11,12,13].

Deamidation of Asn residues is one of the non-enzymatic post-translational modifications of proteins. This reaction is typically caused by a nucleophilic attack of the amide nitrogen of the (*N* + 1) residue to the amide carbon of the Asn side chain, resulting in a tetrahedral intermediate that includes gem-hydroxylamine (Figure 1) [14]. Subsequently, deammoniation proceeds to form a succinimide. Finally, hydrolysis of this succinimide results in the formation of Asp or isoAsp. Asparagine residues that are neutral are converted to acidic or biologically uncommon residues, causing crystallin aggregation [15,16,17,18,19]. Since the reaction with (*N* + 1) residues is the starting point for Asn deamidation, the species of the (*N* + 1) residues affected the reaction rate [20,21,22]. Robinson et al. have reported that the deamidation half-life of the Asn-Gly sequence in pentapeptides is approximately 300-fold smaller than that of the Asn-Ile sequence [21]. In general, the bulkiness of the side chain of the (*N* + 1) residue is believed to decrease the deamidation rate of Asn. However, the deamidation half-lives of the Asn-Phe and Asn-Tyr sequences in pentapeptides are approximately three- or four-fold smaller than those of the Asn-Ile and Asn-Val sequences. Furthermore, the deamidation rate of the Asn-His sequence is more than 20-fold faster than that of the Asn-Ile and Asn-Val sequences. An imidazole ring of the His residue can catalyze Asn deamidation and reduce the activation barrier because the imidazole ring has the ability to mediate proton transfer [23]. However, the benzene ring of the Phe residue does not have this ability. Therefore, the effect of (*N* + 1) residues on the deamidation rate of asparagine residues cannot be explained by their bulkiness alone.

Various reports have demonstrated that deamidation occurs in crystallins and that the occurrence is increased in cataractous lenses [24,25,26,27,28,29]. In particular, the level of deamidation in γS-crystallin was simultaneously reported for all Asn residues [25]. γS-Crystallin has five Asn residues (Asn14, Asn37, Asn53, Asn76, and Asn143). In these residues, Asn14, Asn53, and Asn143 are frequently deamidated; notably, approximately 60% of Asn14 is reported to be deamidated in water-insoluble γS-crystallin extracted from the eye lenses of patients with age-related nuclear cataracts. In contrast, the deamidation ratios of Asn37 and Asn76 are 15% and 8%, respectively, that is, they are infrequently deamidated. The (*N* + 1) residues of the frequently deamidated residue are Phe or Tyr residues, whereas the (*N* + 1) residues of Asn37 and Asn76 are Ser and Asp residues, respectively. Namely, despite the presence of (*N* + 1) aromatic amino acid residues with bulky side chains, Asn14, Asn53, and Asn143 are frequently deamidated. In a previous study, we reported that the activation barrier for deamidation is reduced when the N-terminal side of the main chain conformation is in a specific conformation [30]. However, the effects of the aromatic ring of (*N* + 1) residues are unknown. In this study, we investigated the activation barrier and reaction mechanism of the succinimide formation in the Asn-Phe sequences using quantum chemical calculations. For comparison, the calculations for Asn-Ile sequences were also performed. Ile is neither an aromatic nor a hydrophilic amino acid, and peptide experiments revealed that Ile does not promote Asn deamidation. Therefore, the Asn-Ile sequence is optimal for comparison with the Asn-Phe sequence.

## 2. Results and Discussion

The structures of the model compounds used in this study are shown in Figure 2. The compounds were two dipeptides, asparaginyl phenylalanine and asparaginyl isoleucine capped with acetyl (Ace) and methylamino (Nme) groups on the N- and C-termini, respectively (i.e., CH_3_CO-Asn-Phe-NHCH_3_ and CH_3_CO-Asn-Ile-NHCH_3_). The succinimide formation is presumed to proceed through a gem-hydroxylamine intermediate (Figure 3). Two patterns of side chains of phenylalanine residues were assumed, one close to the reacting atom and the other far away. Therefore, two pathways were examined, i.e., pathways where phenylalanine is adjacent (pathway 1) and phenylalanine is distant (pathway 2). A dihydrogen phosphate (H_2_PO_4_^−^) ion was included as the catalyst in the calculations. A H_2_PO_4_^−^ ion has been proposed to serve as a catalyst for nonenzymatic post-translational modifications in several previous studies [30,31,32,33].

### 2.1. Cyclization Step

To investigate the effects of the (*N* + 1) Phe residue on the reaction mechanisms of Asn deamidation, the optimized geometries in the cyclization step are compared between pathways 1 and 2. The optimized geometries of reactant complex (RC), transition state 1 (TS1), and intermediate 1 (INT1) in the cyclization step for CH_3_CO-Asn-Phe-NHCH_3_ are shown in Figure 4. In the RC of both pathways, the H_2_PO_4_^−^ ion bridged the side chain and main chain of the model compound through hydrogen bond formation (Figure 4A,D). The catalytic ion formed a hydrogen bond with the main chain oxygen on the N-terminal side in pathway 1, but formed a hydrogen bond with the main chain oxygen on the C-terminal side in pathway 2. This catalytic H_2_PO_4_^−^ ion mediated the proton transfers involved in the cyclization step. In TS1, the distances between the main chain amide nitrogen and side chain amide carbon (C-N distance) were 2.15 and 1.94 Å in pathways 1 and 2, respectively (Figure 4B,E). This C-N distance in pathway 1 was similar to those of the previously reported phosphate-catalyzed reactions (2.20–2.23 Å) and carbonate-catalyzed reactions (2.01–2.24 Å) [30]. However, the C-N distance in pathway 2 was shorter. In the acetic acid-catalyzed reaction, the C-N distance (1.62–1.77 Å) was reported to be shorter than those of both pathways in this study [34]. The proton transfer and the nucleophilic attack by the main chain nitrogen were completed, and INT1 was formed. Three hydrogen bonds between the compound and the H_2_PO_4_^−^ were observed in INT1 of pathway 1, while two were observed in INT1 of pathway 2 (Figure 4C,F). In pathway 1, the benzene ring of the Phe residue was maintained close to the amide nitrogen of the Asn side chain throughout the cyclization step. The distance between the closest carbon atom of the benzene ring and the hydrogen atom of the side chain amide was 2.88 Å in TS1 for pathway 1, and the distance between the centroid of the benzene ring and the hydrogen atom of the side chain amide was 3.15 Å. At this distance, the benzene ring was thought to interact with the amide moiety by N-H···π interaction in pathway 1 [35,36]. On the other hand, the benzene ring of the Phe residue remained far from the amide moiety in pathway 2.

The optimized geometries in the cyclization step for CH_3_CO-Asn-Ile-NHCH_3_ are shown in Appendix A. No differences were observed in the reaction mechanism between CH_3_CO-Asn-Phe-NHCH_3_ and CH_3_CO-Asn-Ile-NHCH_3_. In pathway 1 of CH_3_CO-Asn-Ile-NHCH_3_, the interatomic distances and number of hydrogen bonds were similar to those in CH_3_CO-Asn-Phe-NHCH_3_. There were no significant differences between each RC and TS1 in pathway 2 either. In INT1 of CH_3_CO-Asn-Ile-NHCH_3_ in pathway 2 of Ile, three hydrogen bonds were observed, which were similar to INT1 in pathway 1 of CH_3_CO-Asn-Phe-NHCH_3_ and CH_3_CO-Asn-Ile-NHCH_3_. The closest hydrogen atom of the methyl moiety and the nitrogen atom of the amide moiety were 3.01 Å in TS1 for pathway 1, which was a sufficiently interactive distance.

To elucidate the effects of the interaction of the benzene ring, the atomic charge distribution for the benzene ring was investigated (Figure 5). No significant difference was observed in the Mulliken charge for most of the carbon atoms between pathways 1 and 2. However, the charge on one carbon atom was −4.20 e in pathway 1 and −2.66 e in pathway 2. This carbon atom is closest to the side chain amide moiety, and the change in charge distribution is thought to be due to the interaction between the benzene ring and amide moiety.

### 2.2. Deammoniation Step

The deammoniation step had two transition states and proceeded via the two stages in CH_3_CO-Asn-Phe-NHCH_3_. To compare the reaction mechanism in the deammoniation step between pathways 1 and 2, the optimized geometries of intermediate 2 (INT2), transition state 2 (TS2), intermediate 3 (INT3), transition state 3 (TS3), and product complex (PC) in the deammoniation step are shown in Figure 6. Although the position of the H_2_PO_4_^−^ ion and the conformation of the gem-hydroxylamine moiety of INT2 were different from those of INT1, the changes were considered to be easily changed because these ions are abundant under physiological conditions, as H_2_PO_4_^−^ ions are abundant in vivo. In INT2 of both pathways, the H_2_PO_4_^−^ ion formed three hydrogen bonds with the gem-hydroxylamine moiety and the main chain oxygen of the Phe residue (Figure 6A,F). The migrating protons were located between the gem-hydroxylamine moiety and the H_2_PO_4_^−^ ion, indicating that the proton transfer occurs in the first stage (Figure 6B,G). No differences were observed in the distance of each atom between pathways 1 and 2. The proton transfers were completed in INT3, whereas the distances between nitrogen and carbon of the gem-hydroxylamine moiety were 1.59 and 1.63 Å in pathways 1 and 2, respectively, suggesting that there is still an interaction between these atoms (Figure 6C,H). INT3 was directly connected to TS2 and TS3, and the ammonia is released in the second stage. TS3 had the longer distance between nitrogen and carbon of the gem-hydroxylamine moiety (1.97 Å and 1.81 Å in pathways 1 and 2, respectively) in comparison with INT3. In PC, the released ammonia molecule formed a hydrogen bond with the H_2_PO_4_^−^ ion in both pathways, while the other hydrogen bond was formed with the main chain amide hydrogen of the Asn residue only in pathway 2. The H_2_PO_4_^−^ ion formed two hydrogen bonds with the oxygen atoms of the succinimide and the main chain of the Phe residue in both pathways. In a previous study, the deammoniation step progressed in one stage, that is, simultaneous proton transfer and ammonia release [30]. The TS2 reported in the previous study was similar to TS3 of CH_3_CO-Asn-Phe-NHCH_3_ in the present study, and the proton transfers were completed in these TSs. A similar two-stage reaction was also observed in the Maillard reaction, where an amine is added to an aldehyde [37]. Since the same two-stage reaction proceeded in that reverse reaction and deammoniation step, the addition and elimination of amines are considered to likely proceed in two stages.

For CH_3_CO-Asn-Ile-NHCH_3_, the deammoniation step proceeds in one stage (Appendix A), the same as in the previous study [30]. The H_2_PO_4_^−^ ion formed three hydrogen bonds with the gem-hydroxylamine moiety and the main chain oxygen of the Phe residue in INT2. On the other hand, the H_2_PO_4_^−^ ion formed two hydrogen bonds with the gem-hydroxylamine moiety in INT2 of pathway 2, whereas no hydrogen bond was observed between the H_2_PO_4_^−^ ion and the gem-hydroxylamine moiety. Proton transfers and an ammonia release occurred simultaneously, and succinimide was formed. In PC, the H_2_PO_4_^−^ ion formed a hydrogen bond with the main chain oxygen of Phe in pathway 1. In contrast, the ammonia molecule formed a hydrogen bond with the main chain amide hydrogen of Asn in pathway 2. This hydrogen bond formation was similar to that observed for CH_3_CO-Asn-Phe-NHCH_3_. Therefore, there were no differences other than the simultaneous occurrence of proton transfer and deammoniation.

Differences in the atomic charge distribution of the benzene ring were investigated (Figure 7). In TS2, the charge of the δ-position carbon atom was evenly distributed on both sides in pathway 1, but a deviation was observed in pathway 2. Whereas the charge of that carbon atom in TS3 was deviated toward one carbon atom, the deviation was larger in pathway 2 than in pathway 1. In pathway 1, the distance between the centroid of the benzene ring and the closest hydrogen atom of the gem-hydroxylamine was 3.1 Å in TS2 and 3.3 Å in TS3, and the difference in the charge distribution was observed, suggesting that the benzene ring interacted with the amine.

### 2.3. Conformational Change Through the Deamidation

Dihedral angles for all geometries in CH_3_CO-Asn-Phe-NHCH_3_ were retrieved to investigate the conformational changes via the succinimide formation (Table 1 and Table 2). The dihedral angles *φ* and *ψ* characterize the main chain conformation, and *χ* characterizes the side chain conformation (Figure 2). In pathway 1, no significant conformational change of the main chain was observed through the succinimide formation. The largest change of *φ* was 5.0°, which occurred when INT1 was formed from TS1, and that of *ψ* was 8.0°, which occurred when TS1 was formed from RC. In contrast, a large conformational change of the main chain in pathway 2 was observed upon INT3 formation from TS2, and the change in *φ* was 51°. In addition, *ψ* was changed by 41° upon the TS1 formation from RC in pathway 2. Therefore, the deamidation in the Asn-Phe sequence in pathway 2 is considered to proceed with a large structural change of the protein, compared to the reaction in pathway 1. In a previous study, the conformational effect of the main chain on the Asn deamidation was computationally investigated, and the change of dihedral angles in syn conformation was smaller than that in anti-conformation [30]. The largest change in the *φ* and *ψ* values was observed in the TS1 formation for syn conformation, and those values were 13.3° and 31.2°, respectively. In CH_3_CO-Asn-Ile-NHCH_3_, structural changes in the main chain associated with deamidation were observed in both pathway 1 and pathway 2 (Appendix A). The largest change of *ψ* was 38°, which occurred when TS1 was formed from RC in pathway 1. In contrast, a large conformational change of the main chain in pathway 2 was observed upon INT2 formation from INT1, and the change in *φ* was 52°. The structural change in pathway 1 was larger for CH_3_CO-Asn-Ile-NHCH_3_ than for CH_3_CO-Asn-Phe-NHCH_3_. Therefore, the structural changes in the main chain for Asn deamidation in the Asn-Phe sequence are smaller than those in other sequences in pathway 1, while they are larger in pathway 2.

### 2.4. Barrier Heights

To evaluate the differences in activation energy barrier between pathways 1 and 2, the relative energies were calculated using the MP2/6-311+G(2d,2p)//B3LYP/6-31+G(d,p) level, and the energy profiles are shown in Figure 8. In pathway 1 of CH_3_CO-Asn-Phe-NHCH_3_, the barrier height of the cyclization step was higher than that of the deammoniation step (Figure 8A). Therefore, the cyclization step is the rate-determining step in pathway 1. In contrast, the barrier height of the cyclization and deammoniation steps was almost the same as in pathway 2 (Figure 8B). The barrier of the deammoniation step was slightly higher than that of the cyclization step, suggesting that the rate-determining step of pathway 2 was the deammoniation step. However, these barrier heights of pathway 2 were too high for the reaction progress under physiological conditions. A previous study has also reported that the barrier height of the deammoniation step in Gln deamidation with no catalyst or water as a catalyst is higher than or similar to that of the cyclization step, but this alters depending on the catalyst species [38]. The results of this study suggest that the rate-limiting step is also affected by conformation. In pathway 1, the barrier height of the succinimide formation was 67.9 kJ mol^−1^, which was lower than any of the previously reported values by computational studies. The calculated activation barriers of Asn deamidation were 82.7, 84.5, and 113 kJ mol^−1^ [23,30,36]. In Ref. [23], Asn deamidation was investigated in the Asn-His sequence. The adjacent His residue acted as a catalyst, whereas the activation barrier was higher than that in the Asn-Phe sequence in this study. In CH_3_CO-Asn-Ile-NHCH_3_, the rate-determining steps were cyclization and deammoniation steps in pathways 1 and 2, respectively. The barrier heights of both steps in pathway 2 were similar to those for CH_3_CO-Asn-Phe-NHCH_3_. However, the barrier height in pathway 1 was significantly higher than that for CH_3_CO-Asn-Phe-NHCH_3_. Therefore, the decrease in pathway 1 was observed in only CH_3_CO-Asn-Phe-NHCH_3_.

The single-point energies were also calculated using the M06-2X/6-31+G(d,p) level, and the obtained relative energies were similar to those with the MP2/6-311+G(2d,2p)//B3LYP/6-31+G(d,p) (Appendix A). In CH_3_CO-Asn-Phe-NHCH_3_, the barrier heights in the energies obtained using M06-2X/6-31+G(d,p) were 68.0 and 101 kJ mol^−1^ in pathways 1 and 2, respectively. In CH_3_CO-Asn-Phe-NHCH_3_, the barrier heights in the energies obtained using M06-2X/6-31+G(d,p) were 92.2 and 121 kJ mol^−1^ in pathways 1 and 2, respectively. The interaction of the benzene ring of the phenylalanine side chain was thought to reduce the barrier height of Asn deamination. The barrier height for Asn deamidation experimentally investigated using peptides is 80–100 kJ mol^−1^ [14,39]. The calculated values in this study were lower than these. This is thought to be because peptides do not form conformations like the model structure in pathway 1. The barrier height obtained in this study is likely to be applicable only when a similar conformation is formed in protein structures.

## 3. Materials and Methods

The initial structure of pathway 1 for both peptides was constructed based on the 3D structure of Asn14 and Asn53 of γS-crystallin registered in the protein data bank (PDB, PDB ID: 2M3T) [40], and that of pathway 2 was constructed based on the anti-conformation in a previous study [30]. All calculations were performed using Gaussian 16 software [41]. Optimization of energy minima and TS geometries was performed without any constraints by density functional theory (DFT) calculations using the B3LYP/6-31+G(d,p) level of theory [42]. Intrinsic reaction coordinate (IRC) calculations were performed from TSs, followed by full geometry optimizations, to confirm that each TS is connected to energy-minimum geometries. The relative energies reported include the zero-point energies calculated at B3LYP/6-31+G(d,p) level. Furthermore, the single-point energies of all optimized geometries were calculated by the MP2/6-311+G(2d,2p) and M06-2X/6-31+G(d,p) levels. All calculations included hydration effects by employing the polarizable continuum model (PCM). To evaluate the structural changes associated with the reaction, the dihedral angles *φ* (C–N–Cα–C) and *ψ* (N–Cα–C–N), which characterize the main chain conformation, and *χ* (N–Cα–Cβ–Cγ), which characterizes the side chain conformation, were retrieved.

## 4. Conclusions

We investigated the reaction mechanism and the activation barrier of the succinimide formation in the Asn-Phe and Asn-Ile sequences using quantum chemical calculations to elucidate the effects of the (*N* + 1) Phe residue on the Asn deamidation. Our calculation results indicated that the (*N* + 1) Phe residue affected the barrier height of the cyclization step and the reaction mechanism of the deammoniation step. In contrast, although the (*N* + 1) Ile residue was located near the amide moiety of TS1 in pathway 1, the barrier height of pathway 1 was not significantly lower than that of pathway 2. Our calculation results suggest that the (*N* + 1) Phe residue interacts with the amide moiety of the Asn side chain and reduces the barrier height of deamidation. The barrier height for CH_3_CO-Asn-Phe-NHCH_3_ in pathway 1 was lower than the previously reported values determined by experimental methods using peptides. However, the experimental data for peptides were inconsistent with the experimental reports for deamidation in proteins. The bulkiness of (*N* + 1) Phe residue is considered to impair Asn deamidation in peptides, and conformations such as the model structure in pathway 1 are hardly formed in peptides. In contrast, the 3D structure formation in proteins allows the Asn-Phe residues to form conformations such as the model structure in pathway 1, reducing the activation barrier of Asn deamidation. In an experimental structure (PDB ID: 2M3T), the aromatic ring of (*N* + 1) Phe and Tyr residues are oriented toward the Asn residues (Asn14, Asn54, or Asn143), suggesting that the conformation of these Asn and (*N* + 1) residues is similar to the model compound in pathway 1 (Figure 9). These conformations were the same for the other crystal structure (PDB ID: 6FD8) [43]. Therefore, the results of this study can explain the higher deamidation rate for Asn with Phe as the (*N* + 1) residue. In addition, the importance of analyzing deamidation while considering protein structure was demonstrated. In the future study, the effects of other (*N* + 1) residues on Asn deamidation should be analyzed in detail. Because the effects of D3 correction on the barrier heights are unclear [44], accumulation of the results obtained by applying D3 correction is important to advance computational research, including aromatic amino acids. Reaction calculations using HPO_4_^2−^ as a catalyst have been reported for the stereo-inversion of succinimide [45]. The differences in reaction mechanisms and activation barriers due to the catalysts should also be the focus of future research. In addition, the hydration effect using explicit water molecules should also be considered for future study.

## Figures and Tables

**Figure 1 ijms-26-06819-f001:**
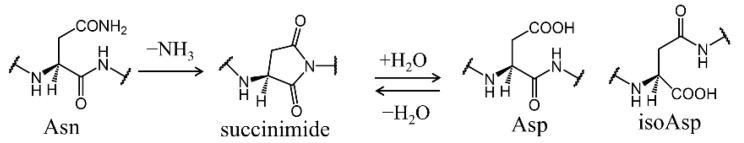
Deamidation pathway of an asparagine residue.

**Figure 2 ijms-26-06819-f002:**
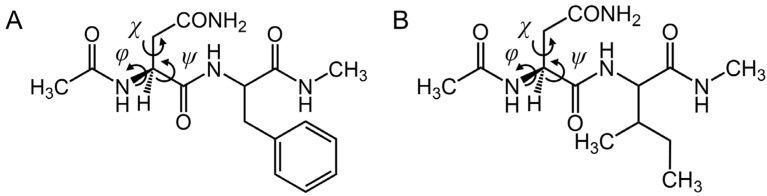
The structures of the model compounds. (**A**) CH_3_CO-Asn-Phe-NHCH_3_ and (**B**) CH_3_CO-Asn-Ile-NHCH_3_. The dihedral angles *φ* (C–N–C_α_–C) and *ψ* (N–C_α_–C–N), which characterize the main chain conformation, and *χ* (N–C_α_–C_β_–C_γ_), which characterizes the side chain conformation.

**Figure 3 ijms-26-06819-f003:**
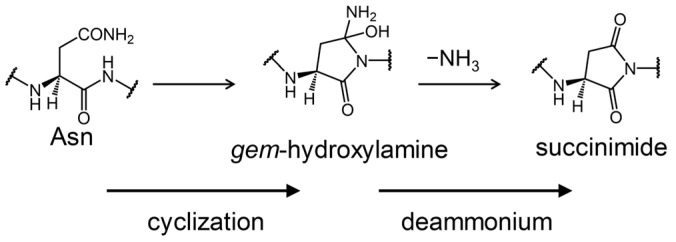
Reaction pathway of the succinimide formation through a gem-hydroxylamine intermediate.

**Figure 4 ijms-26-06819-f004:**
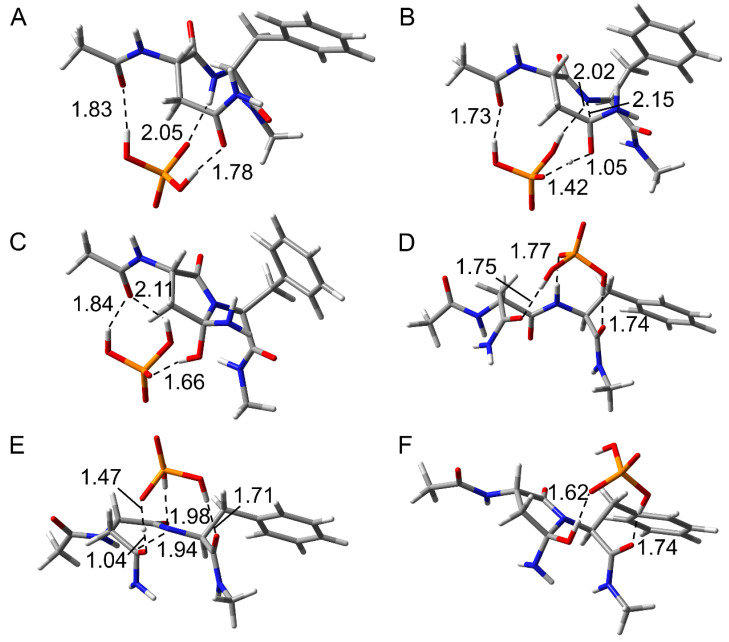
Optimized geometries in the cyclization step for CH_3_CO-Asn-Phe-NHCH_3_. (**A**) RC, (**B**) TS1, and (**C**) INT1 for pathway 1, and (**D**) RC, (**E**) TS1, and (**F**) INT1 for pathway 2 were shown. Carbon, oxygen, nitrogen, phosphorus, and hydrogen atoms were illustrated in gray, red, blue, orange, and white, respectively. Selected interatomic distances are in units of Å.

**Figure 5 ijms-26-06819-f005:**
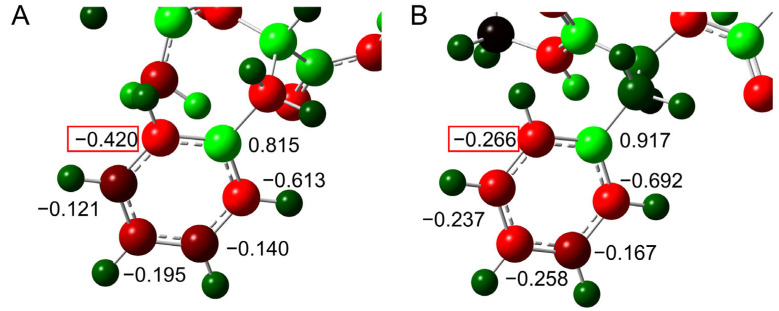
The charges in Coulomb units of carbon atoms in the benzene ring of phenylalanine were calculated by Mulliken methods (/e) with the MP2/6-311+G(2d,2p)//B3LYP/6-31+G(d,p) level. The optimized geometries of TS1 of (**A**) pathway 1 and (**B**) pathway 2 are shown. Relatively negatively charged atoms are shown in red, and relatively positively charged atoms are green. The red boxes indicate the values of δ positions with large differences in the charges.

**Figure 6 ijms-26-06819-f006:**
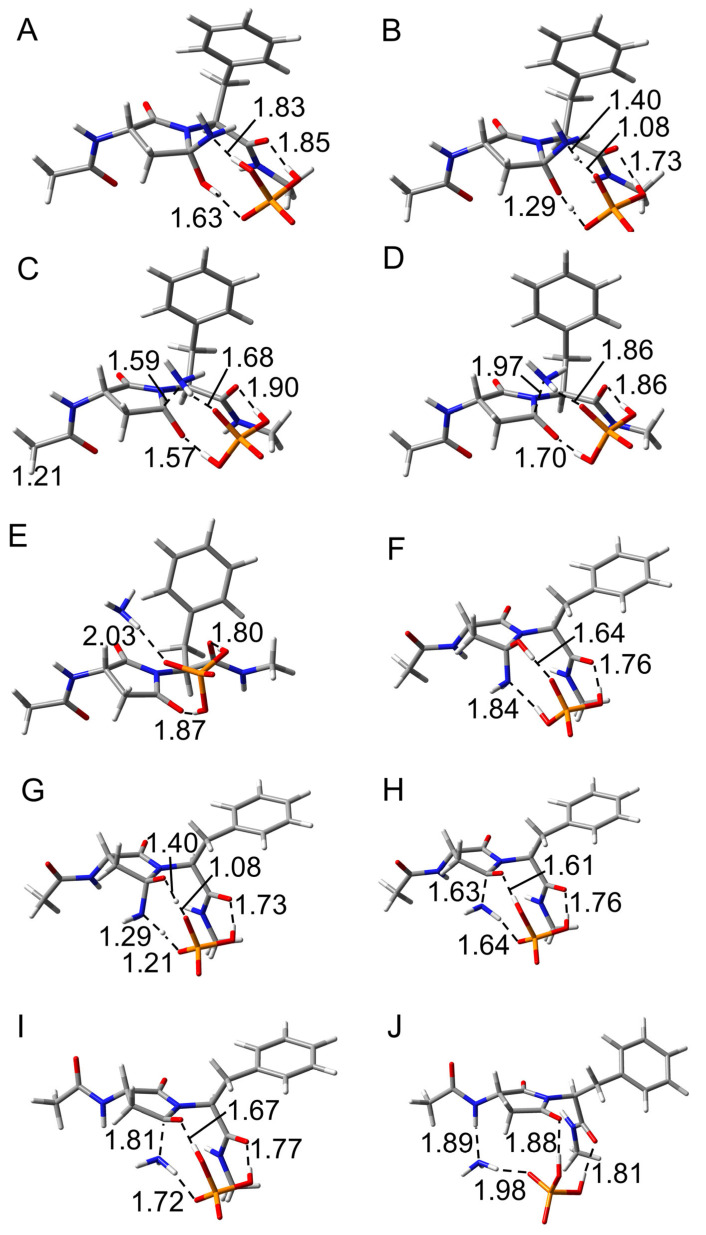
Optimized geometries in the cyclization step for CH_3_CO-Asn-Phe-NHCH_3_. (**A**) INT2, (**B**) TS2, (**C**) INT3, (**D**) TS3, and (**E**) PC for pathway 1, and (**F**) INT2, (**G**) TS2, (**H**) INT3, (**I**) TS3, and (**J**) PC for pathway 2 were shown. Carbon, oxygen, nitrogen, phosphorus, and hydrogen atoms were illustrated in gray, red, blue, orange, and white, respectively. Selected interatomic distances are in units of Å.

**Figure 7 ijms-26-06819-f007:**
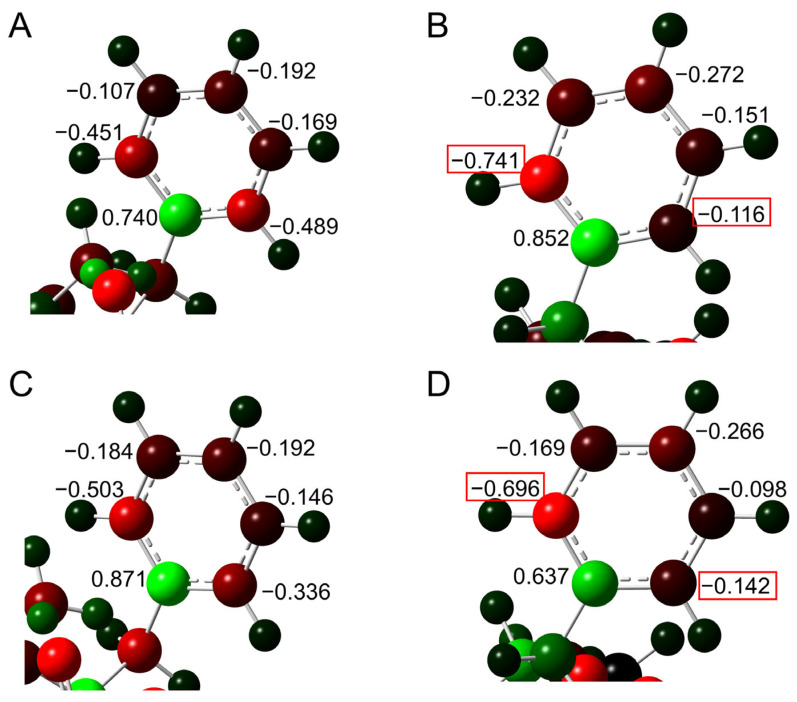
The charges in Coulomb units of carbon atoms in the benzene ring of phenylalanine are calculated by Mulliken methods (/e) with the MP2/6-311+G(2d,2p)//B3LYP/6-31+G(d,p) level. The optimized geometries of (**A**) TS2 in pathway 1, (**B**) TS2 in pathway 2, (**C**) TS3 in pathway 1, and (**D**) TS3 in pathway 2 are shown. Relatively negatively charged atoms are shown in red, and relatively positively charged atoms are green. The red boxes indicated the values of δ positions with large differences in the charges.

**Figure 8 ijms-26-06819-f008:**
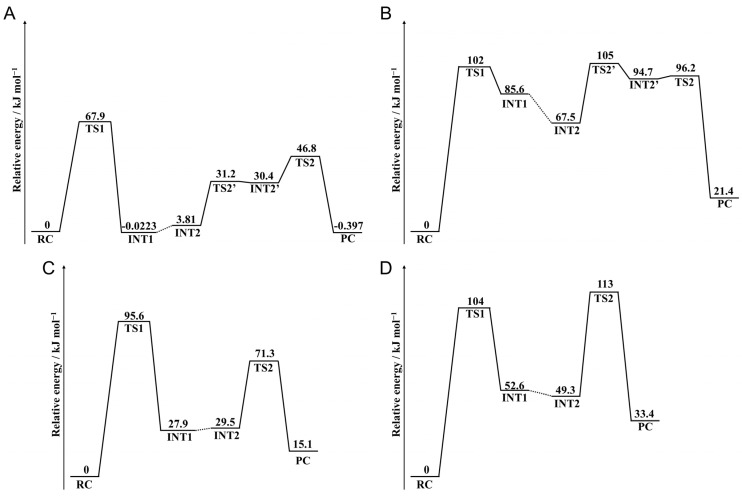
Energy diagrams for the succinimide formation calculated using the MP2/6-311+G(2d,2p)//B3LYP/6-31+G(d,p) level. Entire energy profiles of (**A**) pathway1 and (**B**) pathway 2 in CH_3_CO-Asn-Phe-NHCH_3_ and those of (**C**) pathway1 and (**D**) pathway 2 in CH_3_CO-Asn-Ile-NHCH_3_ are shown.

**Figure 9 ijms-26-06819-f009:**
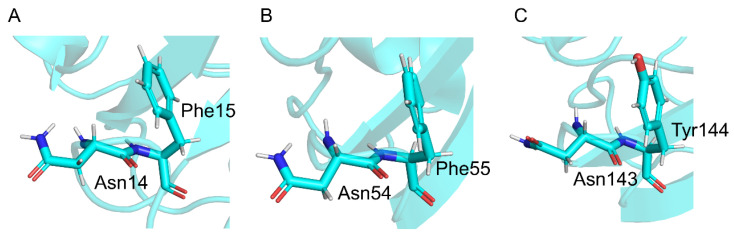
Conformation of the (*N* + 1) aromatic amino acid residues in γS-crystallin. The structures around (**A**) Asn14, (**B**) Asn54, and (**C**) Asn143 were retrieved from the experimentally determined structure (PDB ID: 2M3T). Carbon, nitrogen, oxygen, and hydrogen atoms are shown in cyan, blue, red, and white, respectively.

**Table 1 ijms-26-06819-t001:** Dihedral angles of the optimized geometries in pathway 1.

	Dihedral Angle		
	*φ*	*ψ*	*χ*
RC	61.8	−139	−166
TS1	57.6	−147	166
INT1	52.6	−144	147
INT2	55.8	−146	142
TS2′	55.1	−144	140
INT3	55.6	−144	142
TS2	56.1	−146	144
PC	56.3	−142	136

*φ*: C–N–Cα–C, *ψ*: N–Cα–C–N, and *χ*: N–Cα–Cβ–Cγ.

**Table 2 ijms-26-06819-t002:** Dihedral angles of the optimized geometries in pathway 2.

	Dihedral Angle		
	*φ*	*ψ*	χ
RC	−155	−179	71.8
TS1	−168	−138	98.0
INT1	−166	−132	109
INT2	−162	−112	98.0
TS2′	−167	−108	94.7
INT3	−116	−127	118
TS2	−117	−127	118
PC	−108	−121	117

*φ*: C–N–Cα–C, *ψ*: N–Cα–C–N, and *χ*: N–Cα–Cβ–Cγ.

## Data Availability

Data is contained within the article or the Appendix A. The original contributions presented in this study are included in the article/Appendix A. Further inquiries can be directed to the corresponding author.

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
