# Peer review of "Investigation of the Effect of C-Terminal Adjacent Phenylalanine Residues on Asparagine Deamidation by Quantum Chemical Calculations"

_ijms, 2025, doi:10.3390/ijms26146819_

Round 1

Reviewer 1 Report

Comments and Suggestions for Authors

In this work, the authors explores how the Phe residue adjacent to Asn influences deamidation rates in the human lens protein crystallin. Ab initio calculations revealed that the Phe side chain’s orientation affects the reaction. Using B3LYP and MP2 methods, the authors found that when the Phe benzene ring interacts with the Asn amide group, the activation barrier for succinimide formation decreases, accelerating deamidation. Since structural data shows the aromatic ring facing Asn in crystallin, this interaction likely enhances deamidation, contrasting with prior peptide-based findings.

This work can be interesting for both computational and experimental chemistry community. I would like to ask the authors to address the comments below.

  1. In this work, B3LYP without D3 correction was used. However, the long-range dispersion interaction can be important in the studied system. Can the authors discuss the effect of neglecting the dispersion correction?
  2. The non-covalent interactions between residues are important in the reaction, is there a way to characterize the strength of the non-covalent interaction?
  3. The implicit solvation model was used in this work. If the solvation effect is treated explicitly (adding solvent molecules around), will the results be significantly changed?
  4. The reference for the B3LYP functional is missing: The reference for the B3LYP functional is missing: Phys. Rev. B 37, 785.
  5. page 8, Table 2
    Significant figures in the same column should be consistent.
  6. The unit of Mulliken charge should be e rather than eV.

Author Response

Dear Reviewer:

Thank you very much for your valuable comments. We agree that several points can be clarified and we have modified the article accordingly. Specific responses are provided below. Regarding the correction point of the text is shown in red in the file with marked manuscript revision.

Reviewer1

Comment 1:

In this work, B3LYP without D3 correction was used. However, the long-range dispersion interaction can be important in the studied system. Can the authors discuss the effect of neglecting the dispersion correction?

Response:

As the reviewer mentioned, correction of the long-range dispersion interaction is considered to be important for this studied system. However, comparison of the results with previous studies is essential. Various studies were conducted using B3LYP/6-31+G(d,p) and MP2/6-311+G(2d,2p)//B3LYP/6-31+G(d,p) level. Therefore, we used MP2/6-311+G(2d,2p)//B3LYP/6-31+G(d,p).

Regarding long-range dispersion interactions between aromatic rings, the use of D3 correction may decrease energies of optimized geometries. Tsuzuki et al. have reported that the intermolecular interaction energies calculated for benzene dimer obtained with D3 correction are 3-4 kcal mol-1 lower than those without D3 correction (DOI: 10.5940/jcrsj.61.224). In the systems in this study, effects of long-range dispersion interactions are unclear. If energies of all geometries decrease in the same manner, the relative energies will remain unchanged.

Added sentence to Line 292-294:

Because effects of D3 correction on the barrier heights are unclear [44], accumulation of the results obtained by applying D3 correction is important to advance computational research including aromatic amino acids.

Comment 2:

The non-covalent interactions between residues are important in the reaction, is there a way to characterize the strength of the non-covalent interaction?

Response:

We appreciate the reviewer’s great proposal. To investigate the strength of interactions, extracting fragments or cutting out in clusters like FMO may enable comparing the isolated energy with the energy of each fragment. However, it is unclear which part of the TS should be used as a fragment. Careful consideration may be necessary to determine whether this method can actually be used to investigate the strength of interactions.

Comment 3:

The implicit solvation model was used in this work. If the solvation effect is treated explicitly (adding solvent molecules around), will the results be significantly changed?

Response:

As the reviewer mentioned, using explicit solvent may affect the calculated results. Due to computational cost constraints, it is not practical to treat explicitly all water molecules. However, in a previous study, water molecules have been explicitly with PCM. In that calculation, the activation barrier was approximately 10 kJ mol-1 lower when two water molecules were added compared to when no water molecules were. However, it is necessary to consider how much water is required for each system and whether it is appropriate to use both PCM and explicit water. The conclusion has been amended to include that the hydration effect using explicit water molecules is a topic for future study.

Comment 4:

The reference for the B3LYP functional is missing: The reference for the B3LYP functional is missing: Phys. Rev. B 37, 785.

Response:

We appreciate the reviewer’s pointing out. The reference for the B3LYP functional was added as Ref.42.

Comment 5:

page 8, Table 2

Significant figures in the same column should be consistent.

Response:

All significant figures are aligned to three digits, but if the reviewer meant that they should be aligned horizontally, we have adjusted them.

Comment 6:

The unit of Mulliken charge should be e rather than eV.

Response:

As the reviewer7s suggestion, the unit was modified to e.

Other revision

“B3LYP/6-31+G(d,p)//MP2/6-311+G(2d,2p)” in the captions of Tables S17 and S18 were revised to “MP2/6-311+G(2d,2p)//B3LYP/6-31+G(d,p)”.

We hope that the revised manuscript is acceptable for publication in the International Journal of Molecular Sciences.

Sincerely,

Koichi Kato, Ph. D.

Faculty of Pharmaceutical Sciences, Shonan University of Medical Sciences

16-10 Kamishinano, Totsuka-ku, Yokohama, Kanagawa, 244-0806, Japan

Email Address: kato-k@kinjo-u.ac.jp

Reviewer 2 Report

Comments and Suggestions for Authors

The manuscript by Kato describes a computational study on the Asn deamidation when there is a C-terminal adjacent Phe. This is an interesting problem in protein chemistry, and modeling with small peptide might provide useful insight. However, I find the study less rigorious and complete than the standard of the Journal. The main concerns are:

  1. This redesrch should be conducted by comparison with at least two different peptides. For example the authors use Asn-Phe for the current study. But this should at least be compared with another model peptide such as Asm-Ile or Asn-Val.
  2. The MP2 calculation is not necessarily better than the B3LYP/6-31+G(d,p) in predicting the "barrier heights" or "activation energies", there is no such term as "activation barrier" in chemical dynamics. The M06-2X might even be a better choice to predict the relative energies along the reaction paths.
  3. Figures 6-7 and Tables 1-2 do not help to visualize  the results. 
  4. It is not clear whether the geometry optimization or IRC uses the PCM solvation model or not? The sovent was not specified. The solvation effects need to be specifically discussed, especially when an ionic catalyst is used.
  5.  Line 246, should be "from TSs"? An "f" is missing.
  6. I cannot see how the results justify the conclusion since there is not comparison to other peptide models. 
Comments on the Quality of English Language

The English is fine, minor revision is recommended.

Author Response

Dear Reviewer:

Thank you very much for your valuable comments. We agree that several points can be clarified and we have modified the article accordingly. Specific responses are provided below. Regarding the correction point of the text is shown in red in the file with marked manuscript revision.

Reviewer2

Comment 1:

This redesrch should be conducted by comparison with at least two different peptides. For example the authors use Asn-Phe for the current study. But this should at least be compared with another model peptide such as Asm-Ile or Asn-Val.

Response:

We apologize the lack of explanation. The comparison with other calculation results was described only “which was lower than any of the previously reported values by computational studies.”  The calculated activation barriers of Asn deamidation were 82.7 [Ref. 30], 84.5 [Ref. 23], and 113 [Ref. 36] kJ/mol. In Ref. 23, Asn demidation was investigated in a Asn-His sequence. The adjacent His residue acted as a catalyst, whereas the activation barrier was higher than that in the Asn-Phe sequence in this study (67.9 kJ/mol).  These sentences were added to 2.4 Activation energy.

Although the calculation for Asn demidation in Asn-Ile sequence was performed, the data was included another manuscript currently written. The data is consistent with the conclusions of this paper. However, the results of this paper are consistent with the experimental data (Ref. 25), indicating that the Asn-Ile data is not necessary to be presented as a negative control.

Comment 2:

The MP2 calculation is not necessarily better than the B3LYP/6-31+G(d,p) in predicting the "barrier heights" or "activation energies", there is no such term as "activation barrier" in chemical dynamics. The M06-2X might even be a better choice to predict the relative energies along the reaction paths.

Response:

We appreciate the reviewer’s great proposal. Additional single-point energy calculations were performed. The relative energies obtained by M06-2X/6-31+G(d,p) were similar to those by MP2/6-311+G(2d,2p). Therefore, there was no change in the conclusion. These details have been added to 2.4. Barrier height of activation energy. The calculated energies have been added as Table S19. In addition, “activation barrier” was revised to “barrier height”.

Comment 3:

Figures 6-7 and Tables 1-2 do not help to visualize  the results.

Response:

We agreed with the reviewer’s opinion and revised Figures 6-7. Figure 6 has been enlarged for easier viewing. Figure 7 has been revised to highlight the parts that correspond to the content described in the main text. In addition, “γ-position” in the main text was incorrect and corrected to “δ-position”.

For Table 1-2, the explanation about the dihedral angles were added to Figure 2, Line 200, and Table caption.

Comment 4:

It is not clear whether the geometry optimization or IRC uses the PCM solvation model or not? The sovent was not specified. The solvation effects need to be specifically discussed, especially when an ionic catalyst is used.

Response:

We think the reviewer may have missed that. In Line 267, we described “All calculations included hydration effects by employing the polarizable continuum model (PCM).”

Comment 5:

 Line 246, should be "from TSs"? An "f" is missing.

Response:

We appreciate the pointing out. That was modified to "from TSs".

Comment 6:

I cannot see how the results justify the conclusion since there is not comparison to other peptide models.

Response:

We apologize for the lack of explanation. As described in the response to Comment 1, the comparison with other peptides was added to 2.4. Activation energy. The obtained energy in this study was lower than any of the previously reported values. Therefore, the conclusion is well supported by the results.

Other revision

“B3LYP/6-31+G(d,p)//MP2/6-311+G(2d,2p)” in the captions of Tables S17 and S18 were revised to “MP2/6-311+G(2d,2p)//B3LYP/6-31+G(d,p)”.

We hope that the revised manuscript is acceptable for publication in the International Journal of Molecular Sciences.

Sincerely,

Koichi Kato, Ph. D.

Faculty of Pharmaceutical Sciences, Shonan University of Medical Sciences

16-10 Kamishinano, Totsuka-ku, Yokohama, Kanagawa, 244-0806, Japan

Email Address: kato-k@kinjo-u.ac.jp

Round 2

Reviewer 2 Report

Comments and Suggestions for Authors

The revised manuscript fixed some minor errors and include addition calculation. (M06-2X)

However, the major concerns still exist on the comparison. It is just not scientific sound to compare the current study with previous study using different methods, or with future study.

The authors insisted "The obtained energy in this study was lower than any of the previously reported values. Therefore, the conclusion is well supported by the results." which I cannot agree.

The author also said "Although the calculation for Asn demidation in Asn-Ile sequence was performed, the data was included another manuscript currently written. " In my opinion, this Asn-Ile results should be presented in this manuscript in the first place.

The title of 2.4 should just be "Barrier Heights". Still do they include zero-point energy or thermal energy? 

Comments on the Quality of English Language

Still needs improvement.

Author Response

Dear Reviewer:

Thank you very much for your valuable comments. We agree that several points can be clarified and we have modified the article accordingly. Specific responses are provided below. Regarding the correction point of the text is shown in red in the file with marked manuscript revision.

Comment 1:

It is just not scientific sound to compare the current study with previous study using different methods, or with future study.

The authors insisted "The obtained energy in this study was lower than any of the previously reported values. Therefore, the conclusion is well supported by the results." which I cannot agree.

The author also said "Although the calculation for Asn demidation in Asn-Ile sequence was performed, the data was included another manuscript currently written. " In my opinion, this Asn-Ile results should be presented in this manuscript in the first place.

Response:

As the reviewer mentioned, all calculation for Asn-His dipeptide in Ref. 23 was performed with B3LYP/6-31+G(d,p). Therefore, the results in Asn-Ile sequence were added to the revised manuscript. Because too many data exist, the geometries and dihedral angles of Asn-Ile sequence were shown as Figure S1, Figure S2, Tables S1, and Table S2. In addition, energy diagrams were added to Figure 8. The energy calculation with M06-2X/6-31+G(d,p) were also performed, and the main text was modified throughout. Moreover, the coordinates of the optimized geometries for Asn-Ile sequence were added to SI, and information for Supplementary Materials in the main text was modified.

Comment 2:

The title of 2.4 should just be "Barrier Heights". Still do they include zero-point energy or thermal energy?

Response

The title of 2.4 was revised to "Barrier Heights". We think the reviewer may have missed the latter part. In Line 287, we described “The relative energies for the zero-point energy were corrected by vibrational frequency calculations for all optimized geometries.”

We hope that the revised manuscript is acceptable for publication in the International Journal of Molecular Sciences.

Sincerely,

Koichi Kato, Ph. D.

Faculty of Pharmaceutical Sciences, Shonan University of Medical Sciences

16-10 Kamishinano, Totsuka-ku, Yokohama, Kanagawa, 244-0806, Japan

Email Address: kato-k@kinjo-u.ac.jp

Round 3

Reviewer 2 Report

Comments and Suggestions for Authors

As requested by the editor, I have read the revised manuscript, especially on the addition of Asn-Ile. It makes more sense to have the comparison, and showed the significant difference on the barrier heights on pathway 1. The manuscipt seems to revised in a hurry, so the design and discussion did not fully accomondate the new addition. I suggest the authors elaborate on the comparison. 

The wording in " The relative energies for the zero-point energy were corrected by vibrational frequency calculations for all optimized geometries." is problematic. I think the authors meant "The relative energies reported include the zero-point energies calculated at B3LYP/6-31+G(d,p) level."

Comments on the Quality of English Language

The English still needs refinement throughout the text. For example, line 298: "Our calculation results suggests that" has grammatical error.   

Author Response

Dear Reviewer:

Thank you very much for your valuable comments. We agree that several points can be clarified and we have modified the article accordingly. Specific responses are provided below. Regarding the correction point of the text is shown in red in the file with marked manuscript revision.

Reviewer2

Comment 1:

As requested by the editor, I have read the revised manuscript, especially on the addition of Asn-Ile. It makes more sense to have the comparison, and showed the significant difference on the barrier heights on pathway 1. The manuscipt seems to revised in a hurry, so the design and discussion did not fully accomondate the new addition. I suggest the authors elaborate on the comparison.

Response:

We apologize for giving the impression that we revised in hurry. We added some sentences in Introduction, Discussion, and Conclusion. Please confirm the revised manuscript.

Comment 2:

The wording in " The relative energies for the zero-point energy were corrected by vibrational frequency calculations for all optimized geometries." is problematic. I think the authors meant "The relative energies reported include the zero-point energies calculated at B3LYP/6-31+G(d,p) level."

Response:

As the reviewer suggested, we have revised the sentence.

Other revision

We performed spell checking and proofreading on the entire manuscript.

We hope that the revised manuscript is acceptable for publication in the International Journal of Molecular Sciences.

Sincerely,

Koichi Kato, Ph. D.

Faculty of Pharmaceutical Sciences, Shonan University of Medical Sciences

16-10 Kamishinano, Totsuka-ku, Yokohama, Kanagawa, 244-0806, Japan

Email Address: kato-k@kinjo-u.ac.jp
